

# Effectiveness of adaptive silverware on range of motion of the hand

Susan S. McDonald[1], David Levine[2], Jim Richards[3] and Lauren Aguilar[4]

[1] Department of Occupational Therapy, University of Tennessee—Chattanooga, Chattanooga, TN, United States
[2] Department of Physical Therapy, University of Tennessee—Chattanooga, Chattanooga, TN, United States
[3] Allied Health Research unit, University of Central Lancashire, Preston, United Kingdom
[4] Honors College, University of Tennessee—Chattanooga, Chattanooga, TN, United States

Corresponding author
David Levine, david-levine@utc.edu

## ABSTRACT

**Background.** Hand function is essential to a person's self-efficacy and greatly affects quality of life. Adapted utensils with handles of increased diameters have historically been used to assist individuals with arthritis or other hand disabilities for feeding, and other related activities of daily living. To date, minimal research has examined the biomechanical effects of modified handles, or quantified the differences in ranges of motion (ROM) when using a standard versus a modified handle. The aim of this study was to quantify the ranges of motion (ROM) required for a healthy hand to use different adaptive spoons with electrogoniometry for the purpose of understanding the physiologic advantages that adapted spoons may provide patients with limited ROM.

**Methods.** Hand measurements included the distal interphalangeal joint (DIP), proximal interphalangeal joint (PIP), and metacarpophalangeal joint (MCP) for each finger and the interphalangeal (IP) and MCP joint for the thumb. Participants were 34 females age 18–30 (mean age $20.38 \pm 1.67$) with no previous hand injuries or abnormalities. Participants grasped spoons with standard handles, and spoons with handle diameters of 3.18 cm (1.25 inch), and 4.45 cm (1.75 inch). ROM measurements were obtained with an electrogoniometer to record the angle at each joint for each of the spoon handle sizes.

**Results.** A $3 \times 3 \times 4$ repeated measures ANOVA (Spoon handle size by Joint by Finger) found main effects on ROM of Joint ($F(2, 33) = 318.68$, Partial $\eta^2 = .95$, $p < .001$), Spoon handle size ($F(2, 33) = 598.73$, Partial $\eta^2 = .97$, $p < .001$), and Finger ($F(3, 32) = 163.83$, Partial $\eta^2 = .94$, $p < .001$). As the spoon handle diameter size increased, the range of motion utilized to grasp the spoon handle decreased in all joints and all fingers ($p < 0.01$).

**Discussion.** This study confirms the hypothesis that less range of motion is required to grip utensils with larger diameter handles, which in turn may reduce challenges for patients with limited ROM of the hand.

## INTRODUCTION

Adaptive equipment is used by approximately 23% of older adults in the United States, indicating the importance of validating the efficacy and effectiveness of these assistive devices for optimal and appropriate evidence-based prescription (*Kraskowsky & Finlayson, 2001*). Hand impairment can inhibit or reduce functional ability to perform many activities of daily living such as dressing, bathing, eating, and other self-care. It has been previously reported that the use of traditional utensils to feed oneself can be difficult and/or painful with impaired hand function (*Brach et al., 2002*). Objective assessment of hand joint range of motion (ROM) required for functional activities can be valuable in prescribing adaptive equipment for individuals with impairments. A person with normal hand ROM should not feel discomfort in performing tasks such as gripping a standard sized eating utensil; the same task, however, can be difficult if hand range of motion is limited due to either injury or disability. Examples of conditions that commonly affect hand ROM include stroke, osteoarthritis, rheumatoid arthritis, and cerebral palsy (*Van Roon & Steenbergen, 2006*). According to the *Arthritis Foundation (2015)*, 1 in 5 adults in the United States are affected by arthritis, indicating a great demand for methods to relieve associated complications. A common intervention consists of using increased diameter grip handles on eating utensils. These grips are typically made from a foam-like material and are available in varying sizes such as 3.18 cm (1.25 inch) and 4.45 cm (1.75 inch) diameters as seen in Fig. 1.

Although adaptive utensils with modified handles are commonly used, limited research quantifies the biomechanical effects of larger grips or describes how modified handles affect the ROM of hand joints. Of the prescribed eating and drinking adaptive devices, patients were found to not use 35% of them (*Neville-Jan et al., 1993*). Primary reasons for this noncompliance likely stem from the improper sizing of recommended device (*Kraskowsky & Finlayson, 2001*; *Neville-Jan et al., 1993*). An in-depth review of the literature by *Thomas, Pinkelman & Gardine (2010)* found the four most common reasons for non-compliance for using adaptive equipment are: (1) the patient was not included in deciding on adaptive equipment; (2) inadequate instructions were given; (3) the medical condition improves so they no longer need the adaptive equipment; and (4) the patient's environment is favorable to their condition so they no longer need the adaptive equipment. An individualized approach for prescribing assistive equipment that improves the quality of life for clients mirrors the client-centeredness of rehabilitation therapists. A client-centered approach to assistive equipment provision requires client input when deciding on equipment and to ensure its relevance and appropriateness for the client (*Hoffmann & McKenna, 2004*). Determining the individuals ROM can help with adaptive equipment prescription and may decrease pain associated with simple tasks of daily life and improve utilization and evidence-based rehabilitation outcomes. *Bazanski (2010)* suggested that a 50° lack of flexion in metacarpophalangeal joints, the most important joints during grip, causes a 24% increase in finger impairment.

Electrogoniometers have previously been found to be a valid and reliable tool for the measurement of ROM (*Bronner, Agraharasamakulam & Ojofeitimi, 2010*; *Carnaz et al.,*

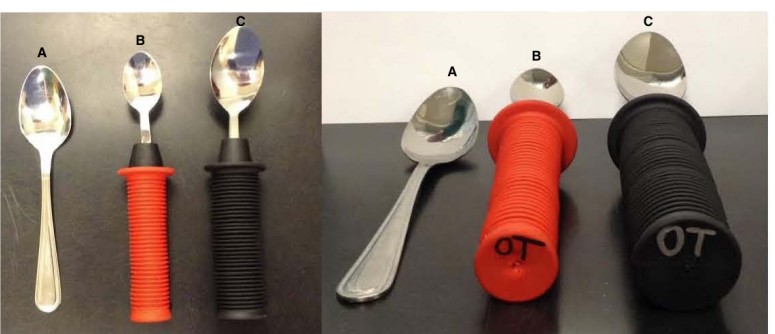

**Figure 1  Adaptive utensils with modified handles.** These images depict a standard spoon (A), a spoon with a 3.18 cm (1.25 inch) diameter handle (B), and a spoon with a 4.45 cm diameter handle (1.75 inch) (C).

*2013*; *Piriyaprasarth et al., 2008*). One previous study used a biaxial goniometer to analyze thumb movements during the use of hand held devices, such as mobile phones, and found the electrogoniometer to be both clinically feasible and accurate (*Jonsson, Johnson & Hagberg, 2007*).

Modified spoon handles can be beneficial while feeding and research has shown positive outcomes regarding the potential benefits of these utensils for patients with conditions including rheumatoid arthritis, Parkinson's disease, and cerebral palsy (*Ma et al., 2008*; *Van Roon & Steenbergen, 2006*). Handle diameter and its relationship to spoon-use movement was examined in patients with Parkinson's disease. Handles of small (1.2 cm), medium (2.0 cm), and large (3.8 cm) diameter size were studied and the large handles significantly decreased task movement time and subjective scores of comfort and feasibility of use (*Ma et al., 2008*). This was likely seen as the hand aperture of the participants with Parkinson's disease was significantly smaller than that of the controls. This study provides evidence of the benefits of altering handle size, but accounts for only the overall movement of the hand as a single unit, and does not address how the grip affects individual joints within the hand.

The use of modified handles for daily activities in persons with rheumatoid arthritis suggests that these assistive devices can help to protect joint integrity by minimizing joint forces and avoiding tight grips (*Shipham & Pitout, 2003*). *Van Roon & Steenbergen (2006)* examined spoon grip-size and its effects on movement kinematics and food spilling for patients with cerebral palsy. Participants with tetraparesis performed quicker transportation of water from one bowl to another and with less spillage when using a 5 cm (2 inch) diameter modified spoon versus a 3 cm (1.18 inch) and 1 cm (0.40 inch) spoon.

While these studies show benefits that may result from using modified spoon handles, they do not study biomechanical changes that occur to individual finger joints when gripping the handles. This study aimed to determine the biomechanical differences in ROM of the fingers when using three different spoon handles in young healthy subjects. These included a standard spoon, a 3.18 cm (1.25 inch) diameter modified handle and a 4.45 cm (1.75 inch) diameter modified handle. These sizes were chosen as they are commonly adopted by patients among those commercially available. The purpose of this

study was to determine differences in the ROM required from the joints in the hand when gripping three different sizes of adaptive spoon handles with various diameters.

## MATERIALS & METHODS

### Subjects

Thirty-four healthy females who were students at the University of Tennessee at Chattanooga, between the ages of 18 and 30 ($x = 20.38 \pm 1.67$) years of age, voluntarily participated in this study. The average grip strength was 58.41 psi, consistent with previously published normative values for females between the ages of 20–29 (*Bohannon, 2006*; *Peters et al., 2011*).

Exclusion criteria included previous hand injury, any neurological condition that would impair hand movement, arthritis or any other condition that would prevent the subject from having normal hand function and ROM. To reduce the amount of variables potentially affecting or influencing results, all participants were right handed and only the dominant sides were assessed, as the dominant hand is typically used to grasp utensils. All subjects read and signed an informed consent form in accordance with the Institutional Review Board at the University of Tennessee at Chattanooga (IRB #14-026). There were no incentives or rewards given for participating. Subjects were recruited using online advertisements sent to students of the University of Tennessee at Chattanooga.

### Equipment

A Jamar hydraulic hand dynamometer (Patterson Medical, Warrenville, IL, USA) was used to take a total of 3 measurements of grip strength, which were averaged. The electrogoniometer utilized (Biometrics Ltd, Ladysmith, VA, USA) was comprised of an angle display unit and a single axis goniometer with accuracy previously reported as $\pm 0.1°$ (*Christensen, 1999*). A foam arm rest (Fig. 2) was used to provide a comfortable standardized position for the subjects during data collection.

### Experimental protocol

Subjects were seated with their shoulder in the anatomical position, and their elbow at a 90° angle, with the hand dynamometer handle placed in the second grip position which is recognized as the standard position for producing the most accurate results (*Massy-Westropp et al., 2011*; *Trampisch et al., 2012*). Grip strength was tested by asking participants to maintain a maximal isometric contraction for 3 s. Participants then placed their right arm on a foam armrest to standardize arm position (Fig. 2). A single axis electrogoniometer was used to measure the angles created at each joint of the hand (Fig. 3).

For all finger joint measurements subjects were given the three spoons (standard handles, and handle diameter of 3.18 cm (1.25 inch), and handle diameter of 4.45 cm (1.75 inch)) in randomized order and instructed to grip the spoon as if they were going to feed themselves while keeping all fingers in contact with the spoon. In order to confirm that the subjects maintained a solid grip on the spoon throughout the experiment, a small lightweight object was placed in the spoon to ensure they could lift and balance an object with their grip. Hand measurements included the distal interphalangeal joint (DIP), proximal interphalangeal

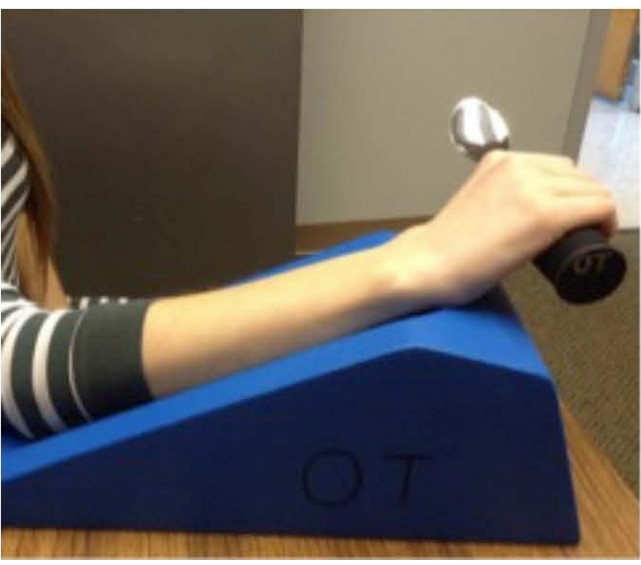

**Figure 2** **Foam arm rest to support the forearm.**

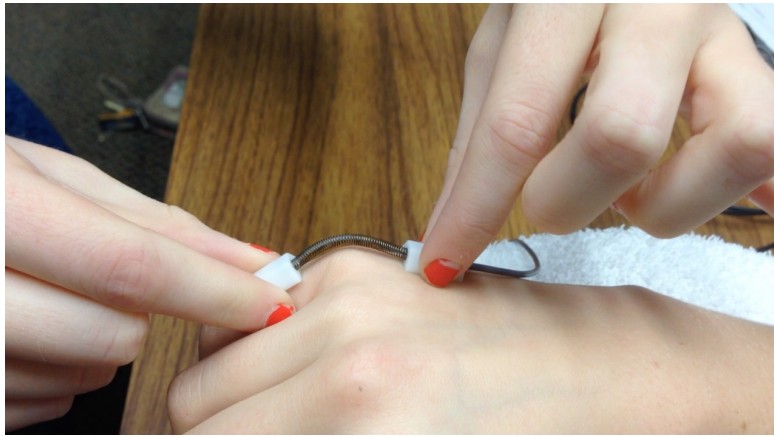

**Figure 3** **Single axis electrogoniometer measuring flexion of the fifth (pinky) finger metacarpopha-langeal (MCP) joint.** Image is demonstrating measuring the MCP joint of the pinky finger. Sensor 'A' is placed on the metacarpal shaft and sensor 'B' is placed on the proximal phalanx. (Source: *Goniometer and Torsiometer Operating Manual.* Biometrics Ltd.)

joint (PIP), and metacarpophalangeal joint (MCP) for each finger and the interphalangeal (IP) and MCP joint for the thumb. Measurements were obtained for all joints and all fingers by placing one sensor on the proximal bone and one sensor on the distal bone adjacent to the joint being measured (Fig. 3 displays an example of the pinky finger MCP). The angle was displayed on the display unit and was recorded. All measurements were made in triplicate.

**Table 1** Comparison of thumb (first digit) ROM using a standard spoon, and two commercial spoons with enlarged diameter handles (3.18 cm (1.25-inch) and 4.45 cm (1.75-inch)).

|  | MCP | IP |
|---|---|---|
| Standard handle | $30.62° \pm 16.08°$ | $45.7° \pm 19.61°$ |
| 3.18 cm (1.25 inch) handle | $26.46° \pm 14.50°$[a] | $42.28° \pm 10.93°$[a] |
| 4.45 cm (1.75 inch) handle | $16.53° \pm 14.57°$[a,b] | $36.43° \pm 12.13°$[a,b] |

**Notes.**
[a] Difference between modified handles and standard handle ($P < 0.01$).
[b] Difference between 3.18 and 4.45 cm handles ($P < 0.01$).

**Table 2** Comparison of index finger (second digit) ROM using a standard spoon, and two commercial spoons with enlarged diameter handles (3.18 cm (1.25-inch) and 4.45 cm (1.75-inch)).

|  | MCP | PIP | DIP |
|---|---|---|---|
| Standard handle | $87.47° \pm 12.12°$ | $106.59° \pm 7.70°$ | $63.58° \pm 11.33°$ |
| 3.18 cm (1.25 inch) handle | $56.98° \pm 13.28°$[a] | $70.73° \pm 6.36°$[a] | $45.86° \pm 6.80°$[a] |
| 4.45 cm (1.75 inch) handle | $40.68° \pm 11.77°$[a,b] | $55.01° \pm 8.13°$[a,b] | $35.59° \pm 6.96°$[a,b] |

**Notes.**
[a] Difference between modified handles and standard handle ($P < 0.01$).
[b] Difference between 3.18 and 4.45 cm handles ($P < 0.01$).

**Table 3** Comparison of middle finger (third digit) ROM using a standard spoon, and two commercial spoons with enlarged diameter handles (3.18 cm (1.25-inch) and 4.45 cm (1.75-inch)).

|  | MCP | PIP | DIP |
|---|---|---|---|
| Standard handle | $93.66° \pm 10.12°$ | $104.53° \pm 5.51°$ | $71.31° \pm 11.01°$ |
| 3.18 cm (1.25 inch) handle | $67.42° \pm 12.89°$[a] | $67.1° \pm 5.78°$[a] | $50.93° \pm 7.07°$[a] |
| 4.45 cm (1.75 inch) handle | $52.98° \pm 12.23°$[a,b] | $53.68° \pm 4.94°$[a,b] | $39.71° \pm 7.43°$[a,b] |

**Notes.**
[a] Difference between modified handles and standard handle ($P < 0.01$).
[b] Difference between 3.18 and 4.45 cm handles ($P < 0.01$).

## RESULTS

Mean values and standard deviations of ROM are reported for each finger, by joint and spoon handle size (Tables 1–5). A $3 \times 3 \times 4$ repeated measures ANOVA (Spoon handle size by Joint by Finger) found main effects on ROM of Joint ($F(2, 33) = 318.68$, Partial $\eta^2 = .95$, $p < .001$), Spoon handle size ($F(2, 33) = 598.73$, Partial $\eta^2 = .97$, $p < .001$), and Finger ($F(3, 32) = 163.83$, Partial $\eta^2 = .94$, $p < .001$). Pairwise comparisons showed that as spoon size increased, the range of motion needed decreased in all joints and all fingers ($p < 0.01$). In all five fingers the differences in ROM between the standard spoon and both adaptive spoons was statistically significant ($p < 0.01$), with the adaptive spoons requiring less ROM for grasp. In all five fingers the difference in ROM between the 3.18 cm (1.25 inch) diameter and 4.45 cm (1.75 inch) diameter spoons was statistically significant ($p < 0.01$) with the 4.45 cm (1.75 inch) diameter spoon requiring less ROM for grasp (Tables 1–5).

**Table 4** Comparison of ring finger (fourth digit) ROM using a standard spoon, and two commercial spoons with enlarged diameter handles (3.18 cm (1.25-inch) and 4.45 cm (1.75-inch)).

|  | MCP | PIP | DIP |
|---|---|---|---|
| Standard handle | 81.07° ± 11.79° | 108.9° ± 5.84° | 68.17° ± 11.66° |
| 3.18 cm (1.25 inch) handle | 54.89° ± 15.09°[a] | 68.05° ± 6.22°[a] | 45.98° ± 6.90°[a] |
| 4.45 cm (1.75 inch) handle | 42.33° ± 14.81°[a,b] | 54.82° ± 7.21°[a,b] | 33.03° ± 5.02°[a,b] |

**Notes.**
[a] Difference between modified handles and standard handle ($P < 0.01$).
[b] Difference between 3.18 and 4.45 cm handles ($P < 0.01$).

**Table 5** Comparison of pinky (fifth digit) ROM using a standard spoon, and two commercial spoons with enlarged diameter handles (3.18 cm (1.25-inch) and 4.45 cm (1.75-inch)).

|  | MCP | PIP | DIP |
|---|---|---|---|
| Standard handle | 77.28° ± 19.23° | 96.08° ± 8.21° | 75.76° ± 11.04° |
| 3.18 cm (1.25 inch) handle | 51.96° ± 20.83°[a] | 51.71° ± 9.39°[a] | 39.73° ± 7.49°[a] |
| 4.45 cm (1.75 inch) handle | 39.06° ± 20.07°[a,b] | 42.7° ± 8.76°[a,b] | 31.28° ± 9.78°[a,b] |

**Notes.**
[a] Difference between modified handles and standard handle ($P < 0.01$).
[b] Difference between 3.18 and 4.45 cm handles ($P < 0.01$).

## DISCUSSION

This study quantified finger and thumb joint ROM needed for healthy adult females to grip a standard spoon and two different adaptive spoon handle sizes. A statistical comparison between the ROM for each finger, for each of the three spoons showed a significant difference between the angles formed at each joint, with respect to the spoon handle size. The angle recorded can be thought of as the distance the joint moved from its original position in order to grasp the spoon handle. Joint angles were greater when subjects gripped the standard spoon handle compared to the handles of the modified spoons. The need for greater ROM with a standard spoon indicates a potential challenge for someone with limited hand ROM to grasp a standard sized spoon handle.

The variability of the data obtained for hand ROM was actually smaller than expected (Tables 1–5). Some variability between individuals was likely due to the variations in which people grasp a utensil despite standardized instructions being given or holding the spoon. The data listed in Tables 1–5 and the statistical analyses confirm less range of motion is required to grip spoons with modified handles. Patients who benefit from the use of such utensils include those diagnosed with conditions that commonly restrict hand ROM, such as patients diagnosed with carpal tunnel, stroke, cerebral palsy, or rheumatoid arthritis (*Van Roon & Steenbergen, 2006*) as well as older adults (*Kraskowsky & Finlayson, 2001*). Knowing the ROM required by the hand to attain certain grasps may help reduce trial-and-error approach and improve the prescription of ADL utensils and could be a clinically relevant consideration for occupational therapists who often fit patients with such assistive devices.

### Future research

The aim of this study was to provide quantifiable data to support the common practice of employing adaptive equipment such as spoons with increased handle diameter to reduce ROM required to grip a standard spoon handle and thereby increase independence with feeding activities of daily living. Although this concept was successfully confirmed, different research hypotheses could be formed and tested using similar methods. For example, information recorded during the data collection process such as measurements of hand size could be investigated to show possible correlations between variables of hand size and the range of motion required to grip the different spoon handle diameters. This would require interpretation of individual results as opposed to the overall group analysis run for this particular study. Advances in biomodeling may present the opportunity to provide custom silverware and other tools based on the individual's hand size, strength, and functional needs. Other variables could be introduced such as questioning the subject for a subjective rating of comfort to establish what may be the ideal handle size as decreased ROM does not necessarily correlate to increased comfort levels or increased efficiency. A more diverse study population including patients with hand deficits likely to use adaptive equipment could be included in future studies. Certain variables such as grip strength may also be a factor in determining the effectiveness of adaptive utensils when the study population has pre-existing hand impairment, as grip strength performance is highly related to the ability of a subject to use their hand functionality

## CONCLUSIONS

The study quantified the hand range of motion needed for adults to use a standard spoon and two commonly available commercial adaptive spoons. It was hypothesized that it would require less range of motion to grip the spoons with modified handles. An electrogoniometer was used to determine range of motion data for 34 healthy subjects. Statistical analysis found significant differences in range of motion requirements between spoon handle sizes and confirmed the hypothesis that less range of motion is required to grip the modified utensils.

### Funding

This study was supported by a Provost student research award from The University of Tennessee at Chattanooga. The funders had no role in study design, data collection and analysis, decision to publish, or preparation of the manuscript.

### Grant Disclosures

The following grant information was disclosed by the authors:
Provost student research award.

### Competing Interests

David Levine is an Academic Editor for PeerJ.

## Author Contributions

- Susan S. McDonald conceived and designed the experiments, performed the experiments, contributed reagents/materials/analysis tools, wrote the paper, prepared figures and/or tables, reviewed drafts of the paper.
- David Levine and Lauren Aguilar conceived and designed the experiments, performed the experiments, analyzed the data, contributed reagents/materials/analysis tools, wrote the paper, prepared figures and/or tables, reviewed drafts of the paper.
- Jim Richards analyzed the data, contributed reagents/materials/analysis tools, wrote the paper, prepared figures and/or tables, reviewed drafts of the paper.

## Human Ethics

The following information was supplied relating to ethical approvals (i.e., approving body and any reference numbers):

Institutional Review Board at the University of Tennessee at Chattanooga (IRB #14-026).

## Data Availability

Raw data can be found in the Supplemental Information.

## Supplemental Information

Supplemental information for this article can be found online at http://dx.doi.org/10.7717/peerj.1667#supplemental-information.

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
