# Peer review of "Effectiveness of adaptive silverware on range of motion of the hand"

_PeerJ, doi:10.7717/peerj.1667_

## Round 0.1 · original submission · Minor Revisions

The manuscript is interesting and well written. Please follow the suggestions by the Reviewer 1 as annotations on the manuscript PDF. Also the comments by Reviewer 2 should be addressed in particular including in the text the F and partial eta squared values.

·

Basic reporting

A few sentences present some ambiguities as they are formulated (see attached notations). All the rest about basic reporting is fine.

Experimental design

All these requirements have been satisfied.

Validity of the findings

All these requirements have been satisfied.

Additional comments

The paper is generally well written and easy to read and understand. I must admit that the paper is original, it is a small but very interesting matter since no one has ever asked whether tools with over-sized handgrip were actually more efficient in hand-deficient subjects. It is one of those little things you always see in front of you but you have bever asked why it is like that. So...it's a smart paper.

Reviewer 2 ·

Basic reporting

No Comments

Experimental design

Methods should be described with sufficient information to be reproducible by another investigator.
There are not informations about the reliability of the measures
No randomization was done for each conditions
The tool used have a poor accuracy for this investigation, are necessary 0.01 degrees in fact the Coefficient of Variation is very higher to discriminate the different conditions

Validity of the findings

The data should be robust, statistically sound, and controlled.
There are not information's about:
- the sample power estimation
- the level of the significant
- the Anova (univariate or repeated of the measures)
- Fisher value
- Partial eta square
- the Coefficient of Variation was higher (Variable)
TABLE 1 (MCP 52<88% - IP 25.8<41.1%),
TABLE 2 (MCP 14<29% - IP 7<15% - DIP 18<19%)
TABLE 3 (MCP 11<23% - IP 9<52% - DIP 14<19%)
TABLE 4 (MCP 15<35% - IP 5<13% - DIP 15<17%)
TABLE 5 (MCP 25<51% - IP 8<21% - DIP 15<31%)

The authors should clarify the reason of this variability of the data

Additional comments

This article showed a interesting scenario, but there are some points to overcome.

---

## Round 0.2 · accepted · Accept

Authors followed all the comments making the manuscript stronger and now accepted for publication.